# Decreased Levels of Soluble Developmental Endothelial Locus-1 Are Associated with Thrombotic Microangiopathy in Pregnancy

**DOI:** 10.3390/ijms241411762

**Published:** 2023-07-21

**Authors:** Gioulia Romanidou, Theocharis G. Konstantinidis, Anastasia-Maria Natsi, Konstantia Kantartzi, Maria Panopoulou, Emmanouil Kontomanolis, Christina Tsigalou, Maria Lambropoulou, Eleni Gavriilaki, Stylianos Panagoutsos, Ploumis Pasadakis, Ioannis Mitroulis

**Affiliations:** 1Department of Nephrology, Democritus University of Thrace, University General Hospital of Alexandroupolis, Dragana Campus, 68100 Alexandroupolis, Greece; dr_giouliarom@yahoo.gr (G.R.); kkantart@med.duth.gr (K.K.); spanagou@med.duth.gr (S.P.); ploumis@med.duth.gr (P.P.); 2General Hospital “Sismanoglio”, Sismanoglou 45, 69133 Komotini, Greece; 3Laboratory of Microbiology, School of Medicine, Democritus University of Thrace, Dragana Campus, 68100 Alexandroupolis, Greece; mpanopou@med.duth.gr (M.P.); xtsigalou@yahoo.gr (C.T.); 4First Department of Internal Medicine, Democritus University of Thrace, Dragana Campus, 68100 Alexandroupolis, Greece; anastasiamaria1997@hotmail.com; 5Department of Obstetrics and Gynecology, Democritus University of Thrace, University General Hospital of Alexandroupolis, Dragana Campus, 68100 Alexandroupolis, Greece; ekontoma@med.duth.gr; 6Laboratory of Histology-Embryology, School of Medicine, Democritus University of Thrace, Dragana Campus, 68100 Alexandroupolis, Greece; mlambro@med.duth.gr; 7Hematology Department-BMT Unit, General Hospital of Thessaloniki George Papanikolaou, 57010 Thessaloniki, Greece; elenicelli@yahoo.gr

**Keywords:** HELLP syndrome, preeclampsia, DEL-1, KIM-1

## Abstract

HELLP (Hemolysis, Elevated Liver enzymes and Low Platelets) syndrome is a life-threatening complication of pregnancy, which is often secondary to preeclampsia. To date, there is no biomarker in clinical use for the early stratification of women with preeclampsia who are under increased risk of HELLP syndrome. Herein, we show that the levels of circulating developmental endothelial locus-1 (DEL-1), which is an extracellular immunomodulatory protein, are decreased in patients with HELLP syndrome compared to preeclampsia. DEL-1 levels are also negatively correlated with the circulating levels of kidney injury molecule-1 (KIM-1), which is a biomarker for disorders associated with kidney damage. Receiver-operating characteristic curve analysis for DEL-1 levels and the DEL-1 to KIM-1 ratio demonstrates that these values could be used as a potential biomarker that distinguishes patients with HELLP syndrome and preeclampsia. Finally, we show that placental endothelial cells are a source for DEL-1, and that the expression of this protein in placenta from patients with HELLP syndrome is minimal. Taken together, this study shows that DEL-1 is downregulated in HELLP syndrome both in the circulation and at the affected placental tissue, suggesting a potential role for this protein as a biomarker, which must be further evaluated.

## 1. Introduction

HELLP syndrome is a life-threatening complication in pregnancy characterized by hemolysis, elevated liver enzymes, and low platelet, and it is associated with increased maternal and neonatal morbidity and mortality [1]. It occurs in less than 1% of pregnancies [1], and it develops as a complication in 10–20% of women with preeclampsia [1]. Preeclampsia, presented as hypertension and proteinuria, is the result of abnormal placentation and endothelial dysfunction that affect several organs and that can be complicated with liver damage and thrombotic microangiopathy (TMA), which are the features of HELLP syndrome [2].

Increased levels of soluble FMS-like tyrosine kinase 1 (sFLT1), which is an anti-angiogenic protein, and decreased levels of the proangiogenic proteins placental growth factor (PlGF) are observed before the onset of preeclampsia, and they are used for the early diagnosis of the disorder [3,4]. This deregulation in the balance of the placenta-derived factor affects the vasculature, such as sFLT1, PIGF, endoglin, and endothelin, and it is a key event in the pathogenesis of both preeclampsia and HELLP syndrome [5]. For instance, induced expression sFlt-1 in pregnant rats has been shown to induce kidney injury [4]. The resulting endothelial cell dysfunction is coupled to inflammation due to the activation of immune cell populations and the complement system [6]. Importantly, complement activation has been proposed to mediate the thrombotic microangiopathy that characterizes HELLP [6,7].

Kidney injury molecule-1 (KIM-1) is a transmembrane glycoprotein that is upregulated in tubular cells upon kidney injury, and its measurement in both urine and blood can serve as a sensitive biomarker [8,9,10]. Since preeclampsia and, particularly, HELLP syndrome are associated with proteinuria and acute kidney injury, we measured KIM-1 levels in the circulation of such patients [2]. Additionally, we evaluated the levels of soluble developmental endothelial locus-1 (DEL-1), an extracellular matrix (ECM) protein produced and released by different cell populations, including endothelial cells and macrophages [11,12], which regulates inflammatory responses via interactions with integrins and phospholipids [13].

Here, we shed light on the role of developmental endothelial locus-1 (DEL-1) on thrombotic microangiopathy in pregnancy. Moreover, we analyze the correlation of DEL-1 with a new biomarker of renal function, KIM-1. Finally, we analyze the usefulness of the DEL-1 to KIM-1 ratio as a potential biomarker that distinguishes patients with HELLP syndrome and preeclampsia.

## 2. Results

The levels of soluble DEL-1 were quantified in the serum of the control women with uncomplicated pregnancy (n = 35), preeclampsia (n = 44), or HELLP syndrome (n = 13). The demographic and clinical characteristics of patients are described in Table 1. The levels of DEL-1 were decreased in the serum of patients with HELLP compared to the two other groups, whereas there was no difference between the control group and the patients with preeclampsia (Figure 1A). On the other hand, there was no difference in the sFLT1/PIGF ratio between patients with preeclampsia and HELLP1, which is used as a marker for the diagnosis of preeclampsia (Figure 1B). These findings suggest that the measurement of soluble DEL-1 levels could be used as a biomarker for the diagnosis of HELLP syndrome. To test the accuracy of DEL-1 measurement in the diagnosis of HELLP in the population of patients with hypertensive complications of pregnancy, we performed a receiver-operating characteristic (ROC) curve analysis. The area under curve (AUC) was 0.8187 ± 0.064 (*p* = 0.0006), and levels <676 pg/mL could be used for the diagnosis of HELLP with a specificity of 78.57% and a sensitivity of 76.92% (likelihood ratio = 3.59) (Figure 1C).

We next measured the blood levels of KIM-1, a marker of kidney injury, and we observed that the levels of KIM-1 were increased from control subjects to patients with preeclampsia to patients with HELLP (Figure 1D). The levels of KIM-1 were inversely correlated with those of DEL-1 (Figure 1E). Since there is an upregulation of KIM-1 levels and a downregulation of DEL-1 in HELLP syndrome compared to preeclampsia, we assessed whether the DEL-1 to KIM-1 ratio could be used to distinguish patients with HELLP syndrome from those with preeclampsia. ROC curve analysis revealed that the specificity and the sensitivity of the values of the DEL-1 to KIM-1 ratio < 263.1 were 88.1% and 76.92%, respectively (AUC = 0.9103 ± 0.039, *p* < 0.0001, likelihood ratio = 6.462) (Figure 1F).

Endothelial cells are a major source of DEL-1 [11]. To test whether the aforementioned decrease in the circulating DEL-1 levels in HELLP syndrome is due decreased production by placental cell populations, we performed immunohistochemical analysis of placental tissue from control subjects and patients with preeclampsia and HELLP syndrome. We observed that placental endothelial cells and perivascular were positive for DEL-1 in the control tissue specimens (Figure 2A, Appendix A). However, endothelial cells in placenta from patients with HELLP syndrome did not express DEL-1 (Figure 2B, Appendix A). Taken together, these data suggest that the decreased expression of DEL-1 by placental endothelial could contribute to the decreased circulating levels of this protein in the blood, even though a systemic effect of the inflammatory environment of HELLP syndrome in the production of DEL-1 by endothelial cells cannot be excluded.

## 3. Discussion

HELLP syndrome is a life-threatening complication of pregnancy, which in the majority of cases develops secondary to preeclampsia [1]. Endothelial dysfunction and damage are the hallmark of hypertensive complications of pregnancy [14]. Endothelial cell activation has been previously associated with increased levels of endothelial cell-derived factors in the circulation, such as thrombomodulin, E-selectin, and von Willebrand factor [15,16]. In this per, we have shown that levels of another endothelial cell-derived factor, DEL-1, were decreased in the circulation of patients with preeclampsia compared to the control, and that they were further decreased in patients with HELLP syndrome. DEL-1 is a 52-kDa glycoprotein consisting of three N-terminal epidermal growth factor (EGF)-like repeats and two C-terminal discoidin I-like domains [13], and it was initially proposed to mediate the adhesion of endothelial cells to ECM through interactions between the RGD motif that is present at the EGF-like repeat and avb3 integrin [17]. Further studies proposed that DEL-1 is an angiogenic protein [18] before the description of its role in leukocyte adhesion to endothelial cells [11]. Since then, several functions have been proposed for DEL-1, including the inhibition of leukocyte recruitment in inflammatory disease models [19,20] and cancer [21], clearance of apoptotic cells [12], induction of myelopoiesis [22], and support of complement-dependent phagocytosis [23]. Interestingly, it has been shown that DEL-1 restrains ischemia-induced neovascularization by modulating inflammation [24], whereas it had a protective role in the progression of both hypertension and cardiac remodeling in a mouse model of angiotensin II-induced hypertension [25]. We have shown that DEL-1 is downregulated in the circulation and the placenta of patients with HELLP syndrome. We have also shown that placental endothelial cells are a source of DEL-1, even though we cannot exclude the possibility that the decreased levels of circulating DEL-1 may be attributed to endothelial cells and to other cell types from different organs. Based on the multiple immunomodulatory functions of DEL-1 and the inflammatory nature of HELLP syndrome, we believe that this downregulation could play a significant (yet still unknown) role in the pathogenesis of HELLP syndrome, affecting placental and/or systemic inflammation.

This decrease in the levels of DEL-1 was coupled with increased levels of KIM-1, which is a protein released upon kidney injury. This enabled the use of the DEL-1 to KIM-1 ration as a possible diagnostic tool for HELLP syndrome. Further prospective studies are needed to assess whether the measurement of these two biomarkers could be used for the early identification of pregnant women with preeclampsia, who are under increased risk for the progression of HELLP syndrome. Whether downregulation of DEL-1 downregulation is a part of the pathogenic mechanism that leads to kidney injury due to enhanced infiltration of effector immune cells is also a matter that needs further investigation.

## 4. Materials and Methods

### 4.1. Study Design

This study included 92 pregnant women (preeclampsia n = 44, HELLP n = 13, and control n = 35), with a mean age of 32.38 ± 1.59 yrs. The diagnosis of the preeclampsia and HELLP syndrome was based on the diagnostic criteria of the International Society for the Study of Hypertension in Pregnancy (ISSHP) [26]. For the diagnosis of HELLP syndrome, the concomitant abnormal values of LDH, liver enzymes, and platelet counts were necessary. Levels of ADAMTS13 were measured and anti-phospholipid screening was performed in all of the samples from patients with HELLP syndrome, and this was conducted in order to exclude thrombotic thrombopenic purpura and anti-phospholipid syndrome as possible alternative diagnoses for thrombotic microangiopathy. Serum and urine samples were collected on admission to the hospital, and they were immediately stored at −80 °C. Placental tissue from 3 patients with HELLP syndrome, 2 women with preeclampsia, and 4 women with uncomplicated pregnancy were also collected. The period of sampling was from June 2014 to May 2020. The study was performed at the University General Hospital of Alexandroupolis and Democritus University of Thrace in Alexandroupolis, Greece. The study was conducted according to the guidelines of the Declaration of Helsinki and approved by Institutional Ethics Review Board of the University Hospital of Alexandroupolis, Greece (Ethics Committee identification code: 941), and informed written consent was acquired from the participants of the study.

### 4.2. Immunoassays

To estimate the renal function, serum creatinine was measured using an automated biochemical analyzer, and the estimated glomerular filtration rate (e-GFR) was calculated using the CKD EPI 2012 formula. Serum Kidney Injury Molecule 1 (KIM-1) was measured by ELISA, using commercially available kits according to the instructions of the manufacturer (Cusabio, Wuhan, China). The sFLT1/PIGF ratio was measured by the fully automated Elecsys^®^ immunoassay on the electrochemiluminescence immunoassay platform (Roche Diagnostics, Mannheim, Germany). Human EGF-like repeat and discoidin I-like domain-containing protein 3 (EDIL3) or DEL-1 (Developmental endothelial locus 1) was measured by ELISA, using commercially available kits according to the manufacturer’s instructions (Cusabio, Wuhan, China). ADAMTS13 activity was determined by Technozym ELISA (Technoclone Herstellung von Diagnostika und Arzneimitteln GmbH 1230 Vienna, Austria).

### 4.3. Immunohistochemical Staining

Serial 4 mm sections of tissue blocks were obtained using a Leica RM2030 automated microtome (Leica Microsystems, Wetzlar, Germany). Human placental tissue slides were incubated at 80 °C for 30 min for deparaffinization, and were then incubated in xylene solution for another 10 min. Descending ethanol solutions (100%, 96%, 70%, 50%) were used for tissue rehydration followed by a 15 min incubation with 0.3% H_2_O_2_ to quench endogenous peroxidase activity. All tissue samples were stained with the peroxidase method (Envision FLEX, Mouse/Rabbit detection System, High pH, DAKO, Carpinteria, CA, USA). Antigen retrieval was performed using EnVision FLEX Target Retrieval Solution High pH (50×) (Catalog No K8004). Slides were incubated with 1:100 diluted primary rabbit anti-DEL-1 antibody (Novus Biologicals, LLC, 10730 E. Briarwood Avenue, Centennial, CO, USA) at 4 °C overnight. In parallel, control slides were incubated with non-immunized rabbit serum (negative control), while a positive control from human gastric carcinoma was always used. Visualization of the antibody antigen complex occurred after a 10-min incubation with the EnVision™ FLEX diaminobenzidine (DAB) chromogen (DM827, DAKO, Carpinteria, CA, USA). Sections were mounted and examined using a Nikon Eclipse 50i microscope (Nikon Instech Co., Ltd., Kawasaki, Japan).

### 4.4. Statistical Analysis

The results are presented as mean ± SEM. A D’Agostino–Pearson normality test was used for the assessment of the distribution of data. Data were then analyzed using one-way ANOVA followed by the Tukey multiple comparison test for data that followed normal distribution, and then the Kruskal–Wallis test followed by Dunn’s multiple comparison test for data that followed a non-parametric distribution. Correlation analysis was performed by the Spearman r test. Statistical analyses were performed using GraphPad Prism 9 software (GraphPad Inc., La Jolla, CA, USA). Statistical significance was set at *p* < 0.05.

## 5. Conclusions

Taken together, we propose that DEL-1 could be a potential prognostic biomarker for HELLP syndrome. Its role in the pathogenesis of the syndrome and/or its potential use as a therapeutic target need to be further investigated in prospective clinical trials.

## Figures and Tables

**Figure 1 ijms-24-11762-f001:**
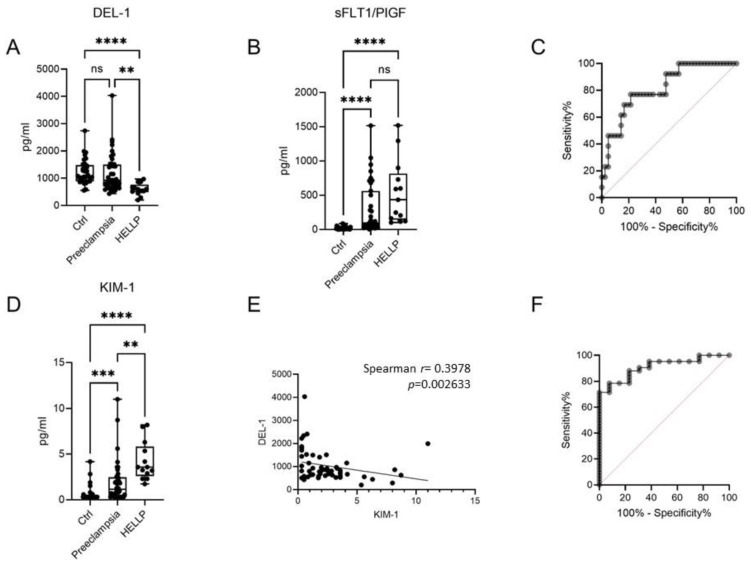
Soluble DEL-1 levels as a biomarker for the diagnosis of HELLP syndrome. (**A**) DEL-1 levels in the serum of women with uncomplicated pregnancies (Ctrl), preeclampsia, and HELLP syndrome. (**B**) The ratio sFLT1/PIGF in the same group of patients. (**C**) ROC analysis showing a significantly higher AUC for DEL-1 levels distinguishing HELLP syndrome from preeclampsia. (**D**) KIM-1 levels in the serum. (**E**) The correlation between DEL-1 levels and KIM-1 levels in patients with preeclampsia and HELLP syndrome. (**F**) ROC analysis showing a significantly higher AUC for DEL-1 to KIM-1 ratio, distinguishing HELLP syndrome from preeclampsia. Kruskal-Wallis test. ** *p* < 0.01, *** *p* < 0.001, **** *p* < 0.0001.

**Figure 2 ijms-24-11762-f002:**
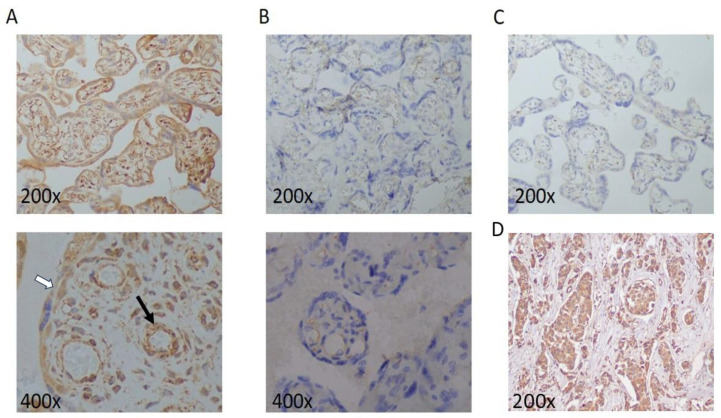
Expression of DEL-1 in placental tissue from patients with HELLP. (**A**) Expression of DEL-1 in tissue sections from a woman with uncomplicated pregnancies assessed by immunohistochemistry, and (**B**) from a patient with HELLP syndrome. Representative data from samples from three individuals in the HELLP syndrome group, two patients with preeclampsia, and four control subjects. (**C**) Technical negative control. (**D**) Positive control––breast adenocarcinoma. (Endothelial cell positive for DEL-1 is indicated with black arrow. Trophoblasts cell positive for DEL-1 are indicated with a white arrow).

**Table 1 ijms-24-11762-t001:** Clinical and demographic characteristics of the study population.

	Preeclampsia *n *= 44	HELLP *n *= 13	Controls *n *= 35	*p* Value
Age (years), mean (±SD),	33.2 ± 2.8	34.6 ± 5.2	29.9 ± 4.3	0.13
Systolic BP,mean (±SD), mm Hg	153.04 ± 7.6	175.3 ± 9.9	128.3 ± 7.1	0.001
Diastolic BP,mean (±SD), mm Hg	88.6 ± 3.4	96.7 ± 8.05	74.7 ± 7.5	0.001
Gestational age (weeks)	34.07 ± 1.8	32.6 ± 3.8	38.09 ± 2.7	0.05
Laboratory findings				
WBC (±SD) K/μl	13,242.6 ± 4915.3	11,963.0 ± 28,635.3	11,237.8 ± 1477.8	0.21
Hb (±SD) g/dL	11.1 ± 0.91	11.7 ± 1.18	12.2 ± 0.5	0.047
Platelets (±SD) K/μl	208.9 ± 24.6	92.3 ± 27.8	231.9 ± 26.9	<0.0001
MPV (±SD)	11.12 ± 0.78	11.6 ± 1.04	8.8 ± 0.82	<0.0001
Glucose (±SD) mg/dL	97.1 ± 12.05	107.1 ± 21.15	95.01 ± 16.3	0.4
eGFR	112.5 ± 12.3	110. ± 23.8	106.3 ± 26.9	0.75
Creatinine mean (±SD) mg/dL	0.68 ± 0.008	0.66 ± 0.013	0.56 ± 0.005	0.64
Urea mean (±SD) mg/dL	23.38 ± 6.7	28.7 ± 13.7	19.72 ± 4.04	0.087
SGOT (±SD) U/L	22.8 ± 8.1	240 ± 98.4	16.4 ± 2.3	<0.0001
SGPT (±SD) U/L	18.8 ± 8.6	182.2 ± 121.4	14.3 ± 3.5	<0.0001
Uric acid (±SD) mg/dL	6.13 ± 0.8	6.8 ± 1.9	4.6 ± 1.04	<0.0001
CRP (±SD) mg/dL	1.5 ± 1.04	1.2 ± 1.4	0.67 ± 0.5	0.028

## Data Availability

The dataset generated during the study contains sensitive personal information. Some of this data is available from the corresponding author on reasonable request.

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
