# Peer review of "Decreased Levels of Soluble Developmental Endothelial Locus-1 Are Associated with Thrombotic Microangiopathy in Pregnancy"

_ijms, 2023, doi:10.3390/ijms241411762_

Round 1
Reviewer 1 Report
Main quesion to the research - possibility to clinical practice?
Topić is very relevant and interesting, but we still need possible to determine high risk group of HEELP.
The article is interesting because it shows a new possibility to isolate a high-risk group of HELLP. There is a very interesting paper, but unfortunately not for clinical usefulness. Many studies should be done at this subject. HELLP is very dangerous complication of pregnancy and we need clearly factors to determine high risk group of this condition.
Three is a not classic form of the paper. There are not conclusions on the end, but there are in the middle of the paper. In my opinion the paper should be reorganised to classic version - introduce, aim, material and method , statistical analysis, dissciusion and conclusions.
The conclusions are consistent with the evidence and arguments.
Three is a not classic form of the paper. There are not conclusions on the end, but there are in the middle of the paper. In my opinion the paper should be reorganised to classic version - introduce, aim, material and method , statistical analysis, dissciusion and conclusions.
Author Response
Thank you for your suggestions and comments, that were all considered valuable and empowered us to improve the quality of our manuscript.
All the corrections in the manuscript are indicated with yellow color.
Main quesion to the research - possibility to clinical practice?
Response
There are no diagnostic markers for accurate diagnosis of HELLP syndrome. The results of the present study were encouraging, nevertheless the new studies that aim to assess the efficacy and diagnostic accuracy of DEL-1/KIM-1 ratio will be processed.
Topić is very relevant and interesting, but we still need possible to determine high risk group of HEELP.
Response
We agree with the reviewer that we still more studies to determine high risk group of HELLP.
The article is interesting because it shows a new possibility to isolate a high-risk group of HELLP. There is a very interesting paper, but unfortunately not for clinical usefulness. Many studies should be done at this subject. HELLP is very dangerous complication of pregnancy, and we need clearly factors to determine high risk group of this condition.
Three is a not classic form of the paper. There are not conclusions on the end, but there are in the middle of the paper. In my opinion the paper should be reorganised to classic version - introduce, aim, material and method, statistical analysis, dissciusion and conclusions.
Response
We apologize for this mistake. We correct all text according to reviewer's comment.
Reviewer 2 Report
The paper "Decreased levels of soluble Developmental endothelial locus-1 are associated with thrombotic microangipathy in pregnancy." reports data of a very interesting study in women with preeclampsia and HELLP syndrome. However there are several points that need to addressed to accept the paper for publicatin:
1) Have the authors evaluated in the women with HELLP the presence of antiphospholipid antibodies? This kind of data should be reported since in this disease is very frequent the positivity for these antibodies especially in case of diffuse microangiopathy.
2) The antiphospholipid antibodies may searched in these cases since the authos should have frozen sera such as reported in the material and methods section. These data should be reported since this may be a relevant to predict the development of HELLP syndrome in pregnant patients.
3) How much time before the development of HELLP syndrome the test for DEL-1 should be performed to predict the HELLP syndrome?
4) Have you evaluated in preeclampsis women whether DEL-1 assessment may be used as an early predictor of hypertensive disorder in pregnancy?
5) The DEL-1/KIM1 ratio may be used for the diagnosis of preeclampsia or HEELP as well as sFLT1/PIGF RATIO? In the result section is not clear, can you specify?
It seems to be quite good
Author Response
Thank you for your suggestions and comments, that were all considered valuable and empowered us to improve the quality of our manuscript.
All the corrections in the manuscript are indicated with yellow color.
The paper "Decreased levels of soluble Developmental endothelial locus-1 are associated with thrombotic microangipathy in pregnancy." reports data of a very interesting study in women with preeclampsia and HELLP syndrome. However there are several points that need to addressed to accept the paper for publicatin:
1) Have the authors evaluated in the women with HELLP the presence of antiphospholipid antibodies? This kind of data should be reported since in this disease is very frequent the positivity for these antibodies especially in case of diffuse microangiopathy.
Response
This is very interesting question from reviewer.
We measurement antiphospholipid antibodies, both ACA and B2GPI, but all samples were negative. The negative results of antiphospholipid antibodies tests were not included in this article.
2) The antiphospholipid antibodies may searched in these cases since the authos should have frozen sera such as reported in the material and methods section. These data should be reported since this may be a relevant to predict the development of HELLP syndrome in pregnant patients.
Response
We measurement antiphospholipid antibodies, both ACA and B2GPI, but all samples were negative. Moreover, special strips that detect b2GPI, Phosphatidylinositol, Phosphatidylserine, Phosphatidylethanolamine, Phosphatidylcholine, Sphingomyelin, were used to determine antiphospholipid antibodies in 18 patients. Due to negative results and cost of reagents the screening with this kit were not runned.
3) How much time before the development of HELLP syndrome the test for DEL-1 should be performed to predict the HELLP syndrome?
4) Have you evaluated in preeclampsis women whether DEL-1 assessment may be used as an early predictor of hypertensive disorder in pregnancy?
Response
It is very interesting question and will be analyzed in next works. Unfortunately, most of our patients were admitted to the hospital as an emergency admission.
5) The DEL-1/KIM1 ratio may be used for the diagnosis of preeclampsia or HEELP as well as sFLT1/PIGF RATIO? In the result section is not clear, can you specify?
Response
The sFLT1/PIGF ratio in used for preeclampsia, but not for HELLP syndrome. We hope that this ratio may be adequate and useful in daily clinical practice